# Trimethylamine N-Oxide Promotes Autoimmunity and a Loss of Vascular Function in Toll-like Receptor 7-Driven Lupus Mice

**DOI:** 10.3390/antiox11010084

**Published:** 2021-12-30

**Authors:** Cristina González-Correa, Javier Moleón, Sofía Miñano, Néstor de la Visitación, Iñaki Robles-Vera, Manuel Gómez-Guzmán, Rosario Jiménez, Miguel Romero, Juan Duarte

**Affiliations:** 1Department of Pharmacology, School of Pharmacy and Center for Biomedical Research (CIBM), University of Granada, 18071 Granada, Spain; cristinagoncor@gmail.com (C.G.-C.); javiermm95@gmail.com (J.M.); sofiaminano@correo.ugr.es (S.M.); nestorvp@correo.ugr.es (N.d.l.V.); irv1991@correo.ugr.es (I.R.-V.); mgguzman@ugr.es (M.G.-G.); miguelr@ugr.es (M.R.); 2Ciber de Enfermedades Cardiovasculares (CIBERCV), 28029 Madrid, Spain; 3Instituto de Investigación Biosanitaria de Granada, ibs.GRANADA, 18007 Granada, Spain

**Keywords:** systemic lupus erythematosus, trimethylamine N-oxide, 3,3-dimethyl-1-butanol, hypertension, cardiovascular complications

## Abstract

Plasma levels of trimethylamine N-oxide (TMAO) are elevated in lupus patients. We analyzed the implication of TMAO in autoimmunity and vascular dysfunction of the murine model of systemic lupus erythematosus (SLE) induced by the activation of the Toll-like receptor (TLR)7 with imiquimod (IMQ). Female BALB/c mice were randomly divided into four groups: untreated control mice, control mice treated with the trimethylamine lyase inhibitor 3,3-dimethyl-1-butanol (DMB), IMQ mice, and IMQ mice treated with DMB. The DMB-treated groups were administered the substance in their drinking water for 8 weeks. Treatment with DMB reduced plasma levels of TMAO in mice with IMQ-induced lupus. DMB prevents the development of hypertension, reduces disease progression (plasma levels of anti-dsDNA autoantibodies, splenomegaly, and proteinuria), reduces polarization of T lymphocytes towards Th17/Th1 in secondary lymph organs, and improves endothelial function in mice with IMQ-induced lupus. The deleterious vascular effects caused by TMAO appear to be associated with an increase in vascular oxidative stress generated by increased NADPH oxidase activity, derived in part from the vascular infiltration of Th17/Th1 lymphocytes, and reduced nrf2-driven antioxidant defense. In conclusion, our findings identified the bacterial-derived TMAO as a regulator of immune system, allowing for the development of autoimmunity and endothelial dysfunction in SLE mice.

## 1. Introduction

Systemic lupus erythematosus (SLE) is a highly deleterious autoimmune inflammatory disease. It can be characterized by the synthesis of autoantibodies, which coalesce into immune complexes, that in turn deposit in target organs, harming the tissues. SLE has been associated with a higher risk for developing renal and cardiovascular disease [1], the most common cause of death in SLE patients [2]. Predominantly young women of child-bearing age are significantly more affected by the pathology. A link between the onset of SLE and the incidence of hypertension has been found [3]. Environmental, genetic, metabolic, and hormonal features partake in SLE predisposition [4]. Nevertheless, the mechanisms behind SLE hypertension are not fully clarified. Inflammatory cytokines and reactive oxygen species (ROS) affect the development of autoimmune diseases-linked pathological rise in blood pressure. These pro-inflammatory elements trigger renal and vascular loss of function and could be downstream of early immune system alterations [5].

Toll-like receptors (TLRs) constitute a class of proteins that act as innate pattern recognition receptors that target a high variety of pathogen-associated molecular patterns (commonly referred to as PAMPs), causing innate immune response mechanisms [6]. Satisfactory evidence has been found to ascertain their participation in the development of human and spontaneous murine models of SLE [7,8]. TLR7 may induce functional and phenotypic variations seen in human SLE, among them, high autoantibody concentrations and multi-organic damage [9]. Moreover, high levels of TLR7 expression, single nucleotide polymorphisms, and upregulation of signaling pathways downstream of TLR7 can be linked to human SLE susceptibility [10]. Recently, it has been shown that TLR7 initiates vascular dysfunction and hypertension in BALB/c mice [11]. This study shows that vascular inflammation and oxidative stress, partly through interleukin (IL)-17 production, are key elements in the development of cardiovascular complications.

Recently, the gut microbiota composition has been associated with the pathogenesis of SLE. Microbial communities in the intestines are able to induce symptoms and generally exacerbate the disease in mouse models of SLE and also patients [12,13,14,15,16,17,18,19,20,21,22,23]. It is also of note that gut dysbiosis was detected in TLR7-dependent murine models of this pathology, and bacterial translocation of *Lactobacillus reuteri* to secondary lymphoid tissues and liver has been proven to drive autoimmunity, being improved by dietary resistant starch, which suppresses the pathological levels of *L. reuteri* and its translocation via short-chain fatty acids (SCFAs) [20]. It has already been shown that gut microbiota, independently of SCFA production, contributes to the TLR7-driven raise in Th17 cells, kidney damage, endothelial dysfunction, vascular inflammation, and hypertension [24], and probiotics consumption decreased BP under TLR7 activation conditions [25]. Diet has been proven the most important element for gut microbiota composition. Choline-, phosphatidylcholine-, and carnitine-rich foods such as liver, eggs, peanuts, and dairy products are partially metabolized into trimethylamine (TMA) by the microbiota and subsequently transformed by the host hepatic flavin monooxygenases into TMA N-oxide (TMAO) [26]. TMAO, as a circulating intestinal microbial metabolite, can trigger vascular inflammation and endothelial dysfunction by formation and activation of nod-like receptor family pyrin domain containing 3 (NLRP3) inflammasomes in endothelial cells [27,28]. In addition, TMAO-induced alloreactive T-cell proliferation and differentiation into Th subtypes in allogenic graft-versus-host disease mice [29]. Interestingly, plasma levels of TMAO were elevated approximately 2.7 times in lupus patients in comparation to healthy individuals [30]. Moreover, we found that TMA-producing bacteria, such as *Desulfovibrio*, are increased in feces from IMQ-treated mice compared to controls, and that its reduction by vancomycin administration improves endothelial function and the raise in BP [24]. Nonetheless, whether TMAO modulates the pathophysiological process of TLR7-driven lupus autoimmunity and its cardiovascular complications is still mostly unknown.

In this study, we tested if TMAO can boost the tendency to SLE development and T cell activation and proliferation in secondary lymph organs, causing endothelial dysfunction and hypertension in a lupus model induced by epicutaneous application of the TLR7 agonist imiquimod (IMQ). To explore the role of this microbial metabolite, we used a structural analog of choline, 3,3-dimethyl-1-butanol (DMB), which nonlethally inhibits TMA formation and reduces the TMAO concentration in mice [31].

## 2. Materials and Methods

### 2.1. Animals and Experimental Groups

The experimental protocols carried out with the animals followed Directive 2010/63/EU of the European Parliament guidelines on the protection of animals utilized for research and NIH guidelines, and received approval from the Ethics Committee of Laboratory Animals of the University of Granada (Spain) (Ref. 12/11/2017/164). Furthermore, our procedures conform to the Transparency on Gut Microbiome Studies in Essential and Experimental Hypertension and the ARRIVE guidelines [32,33,34]. Our group purchased seven- to nine-week-old female BALB/c mice from Janvier (Le Genest, France), which were randomly divided into 4 equally sized experimental groups (*n* = 10): an untreated control (CTR), a control group supplemented with DMB (CTR-DMB), a group treated with imiquimod (IMQ), and an IMQ-treated group supplemented with DMB (IMQ-DMB). All mice were fed a standard chow diet (SAFE A04, Augy, France) and randomized to receive ad libitum access to either normal drinking water (CTR and IMQ groups) or drinking water supplemented with 1% (*v*/*v*, DMB; Sigma-Aldrich, St. Louis, MO, USA). Every day, fresh water bottles were provided. Water and food intake was studied and controlled daily for all groups. A total of 1.25 mg of 5% IMQ cream (Aldara^®^) from Laboratories MEDA PHARMA SALU (Madrid, Spain) was administered through topical application on the right ears of IMQ-treated mice three times per week in alternate days for a total of 8 weeks. Administration of IMQ to the skin effectively triggers the onset of systemic autoimmune disease [9], which justifies the topical use of IMQ to induce an SLE-like autoimmune murine model. Because of their higher predisposition to TLR7-driven functional responses and autoimmunity [35], female experimental subjects were utilized for these experiments.

Our animals were housed in specific pathogen-free (SPF) facilities at the University of Granada Biological Services Unit under standard experimental conditions (12 h light/dark periods, 21–22 °C, 60 ± 10% humidity) in makrolom cages (Ehret, Emmerdingen, Germany) with dust-free laboratory bedding and enrichment. Additionally, the mice were kept in separate cages in order to avoid horizontal transmission of the microbiota. Animals were sorted into groups stochastically, and the experimenters were blinded to drug treatment until data had undergone analysis.

### 2.2. Blood Pressure, Morphology, and Organ Weight Indices

Systolic blood pressure (SBP) was registered in conscious mice using tail-cuff plethysmography (Digital Pressure Meter, LE 5001; Letica S.A., Barcelona, Spain). At least seven values were obtained each session per mouse, and the mean of the lowest three measurements within 5 mmHg was determined to be the SBP [36].

Body weight (in grams) was recorded for all animals. Once the experimental endpoint was reached, all the animals were sacrificed under inhalation anesthesia with isoflurane (2%) and subsequent exsanguination. The hearts with and without the atria and right ventricles were cleaned and weighed in order to assess left ventricle tone. The tibia was dissected in vivo, and the distance between the two tibial epiphyses was measured using a caliper. Organ weight indices were calculated as a ratio with the tibia length for normalization purposes. All organs collected and weighed were then snap-frozen in liquid nitrogen and kept at −80 °C until used for other determinations.

### 2.3. Plasma and Urine Parameters

Blood aliquots were obtained from the left ventricle during the exsanguination, chilled on ice, and centrifuged for 10 min at 3500 rpm at 4 °C. After the centrifuge, the plasma fraction was stored at −80 °C. Plasma anti-ds-DNA antibody activity was quantified through a mouse Anti-dsDNA IgG ELISA Kit (Alpha Diagnostic International, San Antonio, TX, USA) as per the manufacturer’s instructions [37]. Combur Test strips (Roche Diagnostics, Mannheim, Germany) were utilized to determine proteinuria, depositing a drop of instant urine on top of the reactive strip and observing color changes and comparing them with the color guide provided by the manufacturer.

TMAO and TMA level determination in plasma was carried out by stable isotope dilution high-performance liquid chromatography with online electrospray ionization tandem mass spectrometry on an AB SCIEX 5000 triple quadrupole mass spectrometer interfaced with a Shimadzu high-performance liquid chromatography system using a silica column (4.69250 mm, 5 lm Luna Silica; Regis) at a flow rate of 0.8 mL/min. The separation was carried out as described previously [38].

### 2.4. Vascular Reactivity Studies

To determine the endothelial and vascular functions, we used descending thoracic aortic ring-shaped segments in a wire myograph (model 610M, Danish Myo Technology, Aarhus, Denmark) with Krebs solution (in mM: 11 glucose, 118 NaCl, 25 NaHCO_3_, 4.75 KCl, 2 CaCl_2_, 1.2 KH_2_PO_4_, 1.2 MgSO_4_) at 37 °C infused with 95% O_2_ and 5% CO_2_ (pH = 7.4) for isometric tension quantification as extensively detailed in other articles [39]. Length-tension analysis was performed using the Myodaq 2.01 software. Tissue segments were initially tensed to 5 mN for baseline.

In these samples from aorta, concentration-response tests to acetylcholine (ACh, 1 nM–10 μM) were performed, precontracting with the synthetic analog for thromboxane A_2_, U46619 (10 nM). Additional curves to ACh were studied for each sample with the specific pan-NOX inhibitor VAS2870 (10 µM), or the Rho kinase inhibitor Y27632 (0.5 µM) and their controls. The results are shown as a percentage of precontraction tension to U46619.

### 2.5. Measurement of Ex Vivo Vascular Reactive Oxygen Species (ROS) and NADPH Oxidase Activity

Ex vivo vascular ROS were measured by dihydroethidium (DHE, Invitrogen Molecular Probes, Life Technologies S.A. Alcobendas, Madrid, Spain), a red fluorescent dye, which served as a ROS sensor in optimum cutting temperature (OCT) aortic segments, as previously explained [40]. A lucigenin-enhanced assay was used for this determination. Our methodology is extensibly detailed in a previous paper [40]. Briefly, aortic rings were incubated with lucigenin (5 μM) and in the presence or absence of NADPH (100 μM) in order to activate ROS production by the NADPH oxidase in a buffer as follows in mM: NaCl 119, HEPES 20, glucose 5.5, KCl 4.6, CaCl_2_ 1.2, MgSO_4_ 1, NaHCO_3_ 1, Na_2_HPO_4_ 0.15, KH_2_PO_4_ 0.4. The resulting luminescence was recorded for 200 s in a scintillation counter (Lumat LB 9507, Berthold, Germany) in 5 s intervals. Final calculations were performed by subtracting the baseline signal (measures without NADPH) from the ones in the presence of NADPH. Dry tissue weights were estimated for the aortic segments used. We represent NADPH oxidase activity as relative luminescence units (RLU) min^−1^ mg^−1^ dry aortic tissue.

### 2.6. Flow Cytometry

We studied the lymphocyte populations from spleens, mesenteric lymph nodes (MLNs), blood, and aorta through flow cytometry following protocols described in previous publications [23]. Succinctly, we homogenized and filtered the samples through 40 µm cell strainers. Red cells were lysed with Erythrocyte Lysis Buffer. The samples were then incubated with a protein transport inhibitor (BD GolgiPlug^TM^) and stimulated with 50 ng mL^−1^ phorbol 12-myristate 13-acetate plus 1 μg mL^−1^ ionomycin. Next, we blocked the formation of non-specific links with anti-Fc-γ receptor antibodies (Miltenyi Biotec) and incubated with a live/dead viability stain (LIVE/DEAD^®^ Fixable Aqua Dead Cell Stain, Thermo Fisher) for 30 min at 4 °C in PBS. Subsequently, we proceeded to surface staining for 20 min at 4 °C in the dark using the following antibodies: anti-CD45 (FITC, clone 30-F11 Miltenyi), anti-B220 (APC, clone RA3-6B2, BD Bioscience), anti-CD3 (PE, clone REA641 Miltenyi), anti-CD4 (PerCP-Cy5.5, clone RM4-5 Invitrogen), and anti-CD25 (PE-VIO770, clone 7D4 Miltenyi) in flow cytometry staining buffer (FCS; PBS, 1% bovine serum albumin). Cells were then fixed and permeabilized with the buffers A and B sequentially (Fisher Scientific, Madrid, Spain), and intracellular staining was carried out for 30 min at 4 °C in the dark with anti-IL-17a (PE-Cy7, clone eBio17B7, eBioscience, San Diego, CA, USA) and anti-IFN-γ (PE-VIO770, clone XMG1.2, eBioscience, San Diego, CA, USA) for their specific panels. Last, cells were resuspended in test tubes with FCS buffer for acquisition. A Canto II flow cytometer (BD Biosciences) was used to perform the analysis as previously described [39,40]. The gating strategy used was included as Appendix A.

### 2.7. Gene Expression Analysis

The analysis of gene expression was performed by quantitative PCR (qPCR) after the retrotranscription of the RNA present in our samples, as previously described [11]. We homogenized the samples and used TRI Reagent^®^ for total RNA extraction. We quantified RNA levels with the Thermo Scientific NanoDrop™ 2000 Spectrophotometer (Thermo Fisher Scientific, Inc., Waltham, MA, USA), and 2 μg of RNA was retrotranscribed with oligo (dT) primers (Promega, Southampton, UK). The quantitative PCR reactions were conducted using a Techne Techgene thermocycler (Techne, Cambridge, UK). The sequences of the primers utilized for amplification can be found in Appendix A. The efficiency of the PCR reaction was determined using a dilution series of standard vascular samples. Glyceraldehyde-3-phosphate dehydrogenase (GAPDH) was used as a housekeeping gene for normalization. The mRNA relative quantification is represented according to the ΔΔCt method.

### 2.8. Statistical Analysis

Data analysis was performed with GraphPad Prism 8. Result representation follows the structure means ± SEM. Tail SBP and the ex vivo vascular reactivity assays analyses were carried out using two-way analysis of variance (ANOVA) with Tukey’s post hoc method. The rest of the variables were probed for normal distribution with the Shapiro–Wilk normality test and evaluated by one-way ANOVA and Tukey’s post hoc test if a normal distribution was detected, or Mann–Whitney test or Kruskal–Wallis test with Dunn’s multiple comparison test for abnormally distributed data. Statistical significance was considered achieved when *p* < 0.05.

## 3. Results

### 3.1. DMB Prevented High Blood Pressure, Target Organ Damage, and Proteinuria in TLR7-Dependent SLE

Topical administration three times per week in alternate days to wild-type BALB/c mice of IMQ, a TLR7 agonist, successfully induced a lupus-like pathology in our animals not genetically prone to excessive TLR7 signaling [9]. As expected, IMQ-treated mice showed a gradual raise in systolic blood pressure (SBP), being roughly 37 mmHg higher in IMQ than in CTR animals, at the end of the experiment. DMB prevented the development of hypertension induced by IMQ (≈70%) but did not affect CTR group (Figure 1A). We could not detect any significant differences in heart rate (537.7 ± 13.4 bpm vs. 576.0 ± 27.9 bpm, CTR and IMQ groups, respectively), which was unchanged by DMB (Figure 1A). Sustained high BP is one of the most powerful determinants of the development of cardiac and renal hypertrophy. We found that both left ventricular weight/tibia length and kidney weight/tibia length were higher (≈12% and 73%, respectively) in IMQ-treated mice than in control mice. DMB prevented significantly left ventricular hypertrophy, but not renal hyperthophy (Figure 1B). This inducible model is known to present autoimmunity-linked kidney injury [9,11,20]. We found significantly higher protein levels in urine after 4 weeks of IMQ treatment, which was reduced by DMB treatment (Figure 1C), showing prevention of impaired renal function by DMB in IMQ group.

### 3.2. DMB Prevented Disease Activity Progression in TLR7-Dependent SLE

The IMQ model presents increased plasma levels of autoantibodies, splenomegaly, hepatomegaly, and higher type-1 interferon (IFN) expression in lymph organs [9,11]. In the present study, increased plasma levels of anti-dsDNA (Figure 2A), spleen weight/tibia length (Figure 2B), liver weight/tibia length (Figure 2C), and IFNα mRNA levels in MLNs (Figure 2D) were found in IMQ mice. Interestingly, DMB treatment partially prevented the increased plasma anti-dsDNA autoantibodies (≈35%), splenomegaly (≈15%), hepatomegaly (≈23%), and IFNα mRNA levels (≈70%) in IMQ mice. We evaluated the immunomodulatory actions of TLR7 activation by measuring the levels of B cells in spleens and blood from all experimental groups. IMQ treatment led to an increase in the percentages of both splenic and circulating B cells compared with the control group, which were prevented by DMB (Figure 2E). In addition, unchanged total T cell population and lower Th cell proportion were found in the IMQ group as compared to CTR in spleen and blood, which were not affected by DMB treatment (Figure 2E).

### 3.3. Plasma TMAO Increased in TLR7-Dependent SLE and Was Associated with Systolic Blood Pressure and Disease Activity

Fasting plasma concentrations of TMA and TMAO were measured by liquid chromatography–tandem mass spectrometry in all experimental groups. Circulating (plasma) TMA and TMAO concentrations were higher (≈5- and 8-fold, respectively) in IMQ group versus CTR mice (Figure 3A). To investigate whether increases in plasma TMAO concentrations contribute to autoimmunity and high BP in TLR7-dependent systemic autoimmunity, we fed mice a normal chow diet for 8 weeks with their drinking water either supplemented with DMB or not supplemented. DMB reduced plasma TMA (≈43%) and TMAO (≈41%) in IMQ mice but had no effect on plasma levels in CTR animals (Figure 3A). We used regression analysis to determine if SBP or autoimmunity was related to circulating TMAO concentrations. Plasma TMAO level correlated positively with SBP (strong correlation, Pearson *r* = 0.7324) and with plasma anti-dsDNA levels (strong correlation, Pearson *r* = 0.6903) (Figure 3B).

This points to the gut microbiota-produced metabolite TMAO being required for TLR7-dependent systemic autoimmunity—TMAO has a role in the hypertensive effect induced by TLR7 activation. We then studied the mechanisms involved in TMAO-induced, TLR7-dependent hypertension. Taking into account the fact that Th17 infiltration in the vasculature plays a key role in the gut microbiota-mediated changes in BP in TLR 7-driven lupus autoimmunity [24], our group then investigated whether TMAO was able to induce a shift in immune cells in secondary lymph organs.

### 3.4. DMB Treatments Attenuated T-Cell Imbalance

High autoantibody levels and the development of the lupus-like pathology can be linked to a T cell imbalance [11,24,41]. We assessed T cell populations in mesenteric lymph nodes (MLNs), spleens, and blood from our mice. T helper (Th) cell populations (CD3+CD4+) did not present any significant differences among groups in both MLNs and spleen (data not shown). Th17 (CD4+IL-17a+) and Th1 (CD4+IFNγ+) cell relative populations increased in both secondary lymph nodes and in circulation from IMQ mice (Figure 4A–C), whereas T regulatory cells (Treg, CD4+CD25+) were elevated in pathological animals in the spleen and blood (Figure 4B,C). DMB prevented the increase in Th17 and Th1 seen in IMQ in both secondary lymph organs, reducing circulating levels of both type of lymphocytes, but did not seem to affect the proportion of Treg cells. Overall, this suggests that TMAO regulated T-cell polarization, increasing Th17 cell population.

Considering that IL-17a is a crucial component in the mechanisms responsible for endothelial dysfunction in this TLR7-driven lupus autoimmunity model [11], our group then focused on the possible changes induced by DMB on the SLE-linked endothelial dysfunction.

### 3.5. DMB Prevented Endothelial Dysfunction, Vascular Oxidative Stress, and Th17 Vascular Infiltration

Aortae from IMQ group displayed markedly decreased endothelium-dependent vasorelaxation to acetylcholine (ACh) in comparison with CTR (Emax = 59.8 ± 2.7% and 81.7 ± 2.6%, respectively, *p* < 0.01) (Figure 5A). DMB treatment restored this relaxant response (Emax = 79.1 ± 4.2%), being without effect in CTR mice (Emax = 78.2 ± 3.3%). Enhanced aortic ROS content is involved on endothelial dysfunction in SLE [40]. We found a higher (≈62%) red DHE signal in aortic ring from SLE mice, which was abolished by DMB treatment (Figure 5B). When incubated with the NADPH oxidase inhibitor VAS2870, improvement of endothelium-dependent relaxation to ACh was shown in IMQ aortic segments (Figure 5A), which suggests the partial involvement of NADPH oxidase activity in the endothelial dysfunction found in aortic rings from IMQ mice. Furthermore, the NADPH oxidase activity was roughly 1.8-fold higher in IMQ aortic rings than in CTR tissue samples (Figure 5C). Accordingly, mRNA levels of catalytic NOX-4 and regulatory p47^phox^ subunits of NADPH oxidase were also higher in rings from IMQ group in comparison with CTR (Figure 5C). DMB treatment reduced both NADPH oxidase activity and NOX-4 transcript levels, suggesting that TMAO regulated NADPH-driven ROS production to induce endothelial dysfunction. The eNOS inhibitor L-NAME abolished the relaxant response induced by ACh aortic rings from CTR and IMQ groups, involving nitric oxide (NO) in this relaxation [11]. Reduced NO production was detected in aortas from TMAO-supplemented mice, which led to high plasma levels of TMAO (31.5 μM) [28], and by in vitro TMAO incubation (>50 μM). However, we did not find significant changes in eNOS mRNA levels among groups (Figure 5D), suggesting that NO destruction by ROS and reduced NO bioavailability is more relevant to induce endothelial dysfunction than reduced NO production.

Nuclear factor erythroid 2-related factor 2 (NRF2) is a basic leucine zipper transcription factor and is essential for protecting cells against oxidative stress. NRF2 acts through the antioxidant response element (ARE)/electrophile response element (EpRE) to regulate the expression of antioxidative enzymes, such as NADPH:quinone oxidoreductase 1 (NQO1) and heme oxygenase-1 (HO-1), and coordinates a wide range of responses to oxidative damage. A GWAS analysis defined the Nrf2 locus as a region associated with susceptibility to SLE [42]. In addition, Nrf2 activation suppressed lupus nephritis through inhibition of oxidative injury [43]. In aorta from IMQ mice, we found reduced nrf2 mRNA levels, associated with reduced levels of NQO1 and HO-1 (Figure 5E). By contrast, the transcript levels of nrf2 inhibitor keap1 was unchanged. Interestingly, DMB treatment restored nrf2 levels and downstream antioxidant enzymes (Figure 5E), suggesting nfr2 downregulation linked to plasma TMAO levels, which could be partially involved in the endothelial dysfunction induced by IMQ.

Previous studies have shown that TMAO induces vascular inflammation by activating the NLRP3 inflammasome and production of ROS [27,44]. Activation of NLRP3 inflammasome with TMAO is critical for the secretion of IL-1β and ICAM1 gene expression [44]. We found increased mRNA levels of NLRP3 and downstream IL-1β and ICAM1 (Figure 6A) in aorta from IMQ mice as compared to CTR group. Nonetheless, DMB did not affect the expression of the NLRP3 pathway, suggesting that plasma levels of TMAO in IMQ mice were insufficient to directly activate the NLRP3 pathway.

Vascular infiltration of Th17 is an important element to consider when looking for subjacent mechanisms responsible for the endothelial dysfunction triggered by IMQ microbiota [23]. We found higher macrophage and Th17 accumulation in aorta from IMQ than in CTR mice, which were decreased by DMB treatment (Figure 6B), suggesting that TMAO increased vascular immune cells infiltration. The proinflammatory cytokine IL-17 causes Rho-kinase-mediated endothelial dysfunction [23,24], at least in part by NADPH oxidase activation [45] and nrf2 inhibition [46]. The ACh-induced response was also ameliorated in IMQ-treated animals after Y27632 incubation (a Rho kinase inhibitor) (Figure 6C), pointing to Rho kinase activation as a cause for reduced relaxation.

Overall, our data suggest that endothelial dysfunction induced by TMAO under TLR7 activation is mainly mediated by vascular Th17 infiltration, and the subsequent ROS levels increased, linked to increased NADPH oxidase-driven ROS production and reduced nrf2 antioxidant defense. These vascular changes seem to be produced by IL17/Rho kinase pathway activation.

## 4. Discussion

It has been previously demonstrated that gut microbiota has a crucial role in vascular complications associated with SLE [23,24]. Microbial metabolites, the gut microorganism-produced repertoire, mainly include SCFA, TMAO, lipopolysaccharide, H_2_S, uremic toxins, and bile acids [47]. In the present study, we demonstrated the importance of the gut microbiota metabolite TMAO as a regulating agent for autoimmunity, endothelial function, and BP in a TLR-7 activation-induced mouse SLE model. This model simulates more than 80% SLE patients who have a high IFN signature. This is mainly supported by the reduction in plasma anti-dsDNA levels, the improvement of endothelial-mediated vasorelaxation, and the SBP reduction induced by DMB supplementation to IMQ-treated mice. Growing evidence indicates that living microbial therapy, including fecal microbiota transplantation (FMT) [24] and probiotics [25], might be able to improve the microbial dysbiosis and to reduce SLE cardiovascular complications. However, the translation of these strategies that involve fecal transplantation into the clinic, such as FTM from healthy human to SLE patients, is in need of more research. In spite of FMT gaining interest as a therapeutic alternative for autoimmune diseases, their use in clinical practice might be scarce due to practical objections in these chronic pathologies. Moreover, FMT from CTR to IMQ-treated mice decreased vascular alterations but was not able to affect autoimmunity [24]. In our experiment, DMB was able to reduce disease activity and BP, suggesting it as a potential therapeutic drug for SLE.

Plasma TMAO levels were higher in SLE patients as compared to healthy human [30]. We showed for the first time that plasma TMAO levels increased after TLR7 activation, and gut bacterial TMA was inhibited with DMB-reduced plasma TMAO levels. Interestingly, this reduction led to lower autoimmunity, since lower plasma anti-dsDNA levels, splenomegaly, and hepatomegaly were found in IMQ-DMB group as compared to IMQ mice. The mechanisms involved in the progression of SLE activity induced by TMAO are unknown. Humoral immune system components play a crucial role in lupus onset, as shown by evidence suggesting that B cell populations, which are differentiated into antibody-producing plasma cells, are higher in SLE [48]. Accordingly, we found a higher number of B cells in spleen and blood in IMQ mice than in the CTR group, which was normalized with DMB treatment. It is possible that TMAO sensitizes B cells to the binding of antigens and the activation of intracellular transduction pathways, such as protein kinase C (PKC), which lead to the differentiation of B lymphocytes [49]. In fact, TMAO elevated PKC activity in a dose-dependent manner in endotelial cells [50]. Furthermore, in spite of the high proportion of Tregs in lupus mice, these cells cannot control the cumulative impact of multiple genetic triggers of lymphocyte activation and autoreactivity [51]. However, DMB was unable to reduce circulating Treg cell counts, ruling out a possible role of Tregs to lessen autoantibody levels. The administration of a mouse anti-CD20 antibody to deplete B cells ostensibly mitigates autoantibody production and prevents the development of high blood pressure in female NZBWF1 mice [52]. Thus, decreased anti-dsDNA antibody concentration by B cell depletion and lower activation levels may be linked to the antihypertensive effects of DMB supplementation. Additionally, an imbalance between anti-inflammatory Treg and pro-inflammatory Th17 cells is generally seen as a relevant element in both human SLE and murine lupus [53]. We found increased Th17 population in secondary lymph organs from IMQ mice and lower Th17 count induced by TMAO reduction with DMB supplementation, which might be involved in lower autoimmunity induced by DMB. Our results in SLE disease are in agreement to that previously reported, indicating that TMAO control macrophage M1 polarization by NLRP3 inflammasome activation, which provide the cytokines microenvironment in which naïve CD4+ T cells subsequently differentiate into Th1 and Th17 subsets [29]. However, whether this polarization towards Th1 and Th17 is mediated by the polarized M1 macrophage requiring NLRP3 inflammasome activation in our experimental conditions is unknown.

Hypertension can be linked to impaired endothelial function. In recent years, impaired endothelium-dependent relaxation responses to acetylcholine in aortas from TLR7-activated mice have been demonstrated repeatedly [11,24,54]. High vascular ROS levels might be associated with the damage to the endothelial function and the raise in BP in female IMQ-treated animals [11]. In accordance with this, in our experiments, we also detected a poor acetylcholine-dependent relaxation and high ROS content linked to increased NADPH oxidase activity and reduced nrf2-antioxidant defense in IMQ aortic segments when compared to CTR. Interestingly, DMB suppressed NADPH oxidase over-activity and normalized nrf2 pathway, improving endothelial dysfunction. High TMAO concentration induced loss of Ach-induced relaxation. Moreover, treatment with DMB for 8 to 10 weeks to suppress TMA selectively ameliorated endothelium-dependent dilation in old mice to young levels by normalizing vascular superoxide production, improving nitric oxide-mediated dilation, and impeding superoxide-related blocking of endothelium-dependent dilation [28]. Moreover, high TMAO concentrations can induce oxidative stress in cultured endothelial cells via activation of the NLRP3 inflammasome. However, no significant changes in NLRP3 pathway were induced by DMB, suggesting that plasma TMAO levels found in IMQ mice were not sufficient to induce NLRP3 inflammasome at the vascular tissue.

Th17 polarization in secondary lymph organs and Th17 infiltration in aorta plays a crucial part in the vascular alterations caused via TLR-7 activation. In fact, FMT from IMQ mice to control mice caused an increase in plasma IL-17, which led to endothelial function impairment, since IL-17 neutralization restored endothelium-dependent relaxation [23]. We found both reduced Th17 polarization in lymph nodes and vascular infiltration after DMB supplementation linked to improvement of endothelial dysfunction. The pro-inflammatory cytokine IL-17 has been shown to cause be detrimental to the endothelial function, activating the Rho kinase in the vascular wall [55], at least partially, increasing NADPH oxidase-generated ROS [56] and reducing nrf2 pathway [46]. In addition to this, the Rho-kinase inhibitor Y27632 restored the response to acetylcholine in IMQ mice similar to CTR. Targeting the IL-17/IL-17 receptor pathway could be an innovative and effective therapeutic approach for Th17-induced hypertension in SLE patients. However, a few critical but severe adverse events for IL-17-blocking strategies, such as bacterial infections, mucocutaneous candidiasis, and neutropenia are known to appear. In our experiment, DMB, acting in gut microbiota, reduced this pathway, suggesting new pathways to the prevention of SLE-associated cardiovascular complications. However, caution is imperative to extrapolate these findings to humans because of all the potential dissimilitude in the features of the animal and human gut microbiota.

## 5. Conclusions

These experiments show that the gut microbiome metabolic product TMAO increases TLR7 induced autoimmunity and vascular dysfunction. This can be linked to increased B cells differentiation and activation of pro-inflammatory Th17 lymphocytes. Our study provides the missing link between bacterial metabolite derived of choline and lupus severity, revealing the potential to alleviate cardiovascular complications by blockage of the TMAO pathway.

## Figures and Tables

**Figure 1 antioxidants-11-00084-f001:**
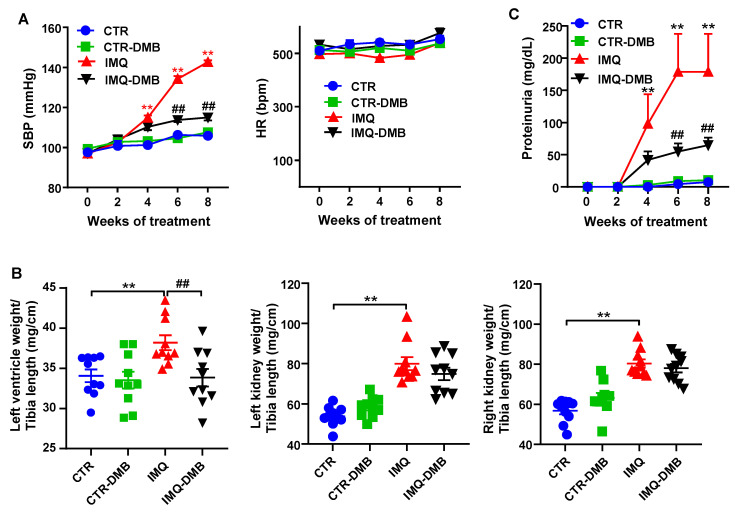
Effects of 3,3-dimethyl-1-butanol (DMB) treatment on blood pressure, organ morphology, and proteinuria in control (CTR) and imiquimod (IMQ) mice. (**A**) Systolic blood pressure (SBP) as determined by tail-cuff plethysmography; (**B**) morphology data as organ weigh/tibia length ratios; and (**C**) proteinuria in CTR, CTR-group treated with DMB (CTR-DMB), IMQ, and IMQ-group treated with DMB (IMQ-DMB). The data are expressed in a means ± SEM format. Tail SBP and proteinuria were tested by two-way ANOVA with the Tukey’s multiple comparison test. The morphological variables were tested with one-way ANOVA and Tukey’s post hoc test, or Kruskal–Wallis test with Dunn’s multiple comparisons. ** *p* < 0.01 in comparison with CTR; **^##^**
*p* < 0.01 in comparison with IMQ.

**Figure 2 antioxidants-11-00084-f002:**
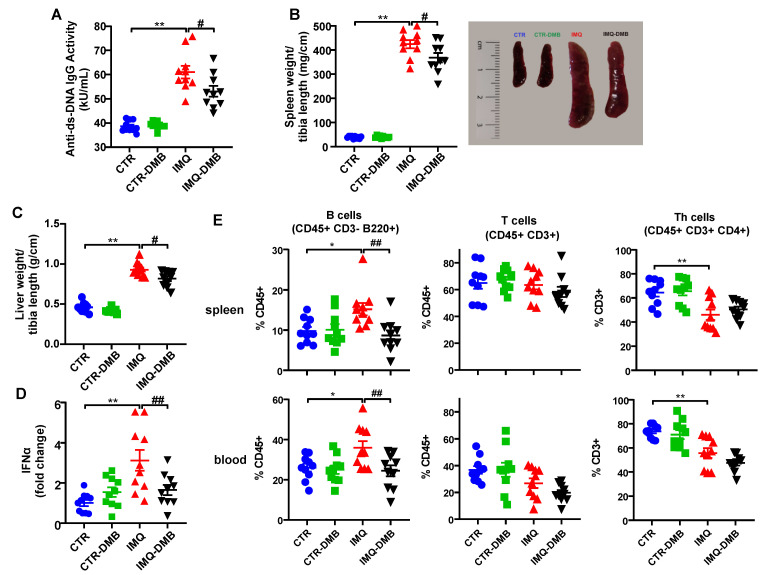
Effects of 3,3-dimethyl-1-butanol (DMB) treatment on disease activity signs in the imiquimod (IMQ) group. (**A**) Circulating double-stranded DNA (anti-ds-DNA); (**B**) splenomegaly; (**C**) hepatomegaly; (**D**) IFNα mRNA levels in MLNs; and (**E**) percentage of B, T, and Th cells in spleen and blood in CTR, CTR-group treated with DMB (CTR-DMB), IMQ, and IMQ-group treated with DMB (IMQ-DMB). The data are expressed in a means ± SEM format. Comparisons between variables were performed with one-way ANOVA and Tukey’s post hoc test, or Kruskal–Wallis test with Dunn’s multiple comparisons. * *p* < 0.05 and ** *p* < 0.01 in comparison with CTR; ^#^ *p* < 0.05 and **^##^***p* < 0.01 in comparison with IMQ.

**Figure 3 antioxidants-11-00084-f003:**
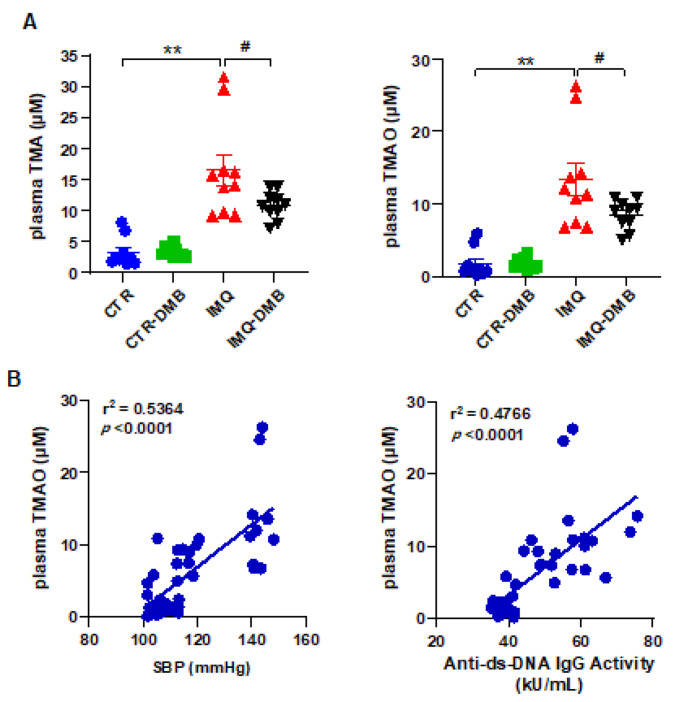
Plasma TMAO was correlated to systolic blood pressure and SLE disease activity. (**A**) Plasma TMA and TMAO levels in control (CTR), CTR-group treated with 3,3-dimethyl-1-butanol (DMB) (CTR-DMB), IMQ, and IMQ-group treated with DMB (IMQ-DMB). The data are represented as means ±SEM. One-way ANOVA and Tukey’s multiple comparison test were performed. ** *p* < 0.01 when compared to CTR; ^#^ *p* < 0.05 when compared to IMQ. (**B**) Pearson correlation between plasma TMAO and SBP or anti-ds-DNA activity using data from all experimental groups.

**Figure 4 antioxidants-11-00084-f004:**
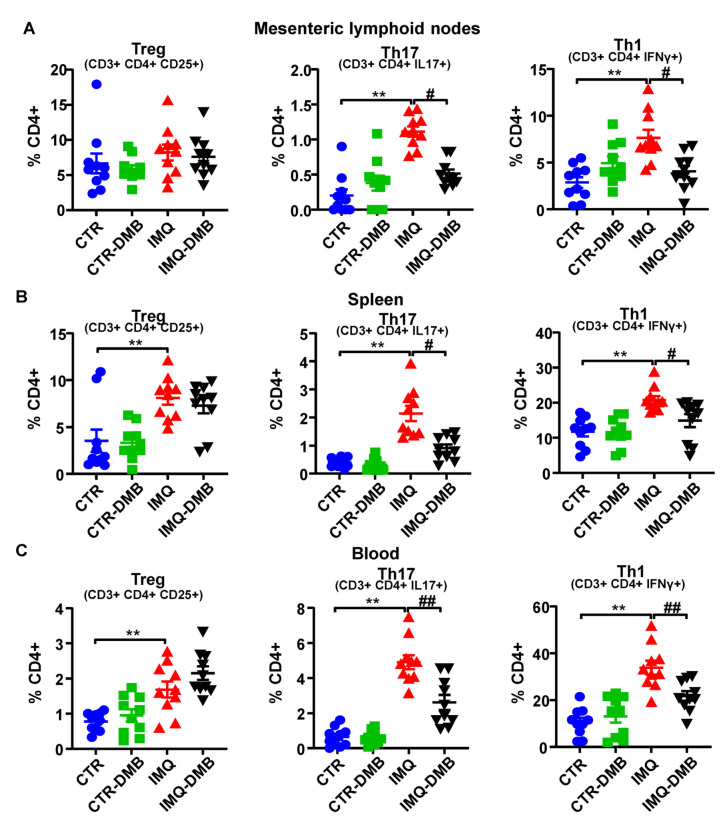
TMAO facilitated lymphocyte activation and proliferation in imiquimod (IMQ) animals. (**A**) Regulatory T (Treg), Th17, and Th1 cells as analyzed through flow cytometry in mesenteric lymph nodes, (**B**) in spleen, and (**C**) blood in CTR, CTR-group treated with DMB (CTR-DMB), IMQ, and IMQ-group treated with DMB (IMQ-DMB). Data are represented as means ± SEM. One-way ANOVA and Tukey’s post hoc test or Kruskal–Wallis test with Dunn’s multiple comparisons were performed. ** *p* < 0.01 compared to CTR; ^#^ *p* < 0.05 and **^##^**
*p* < 0.01 compared to IMQ.

**Figure 5 antioxidants-11-00084-f005:**
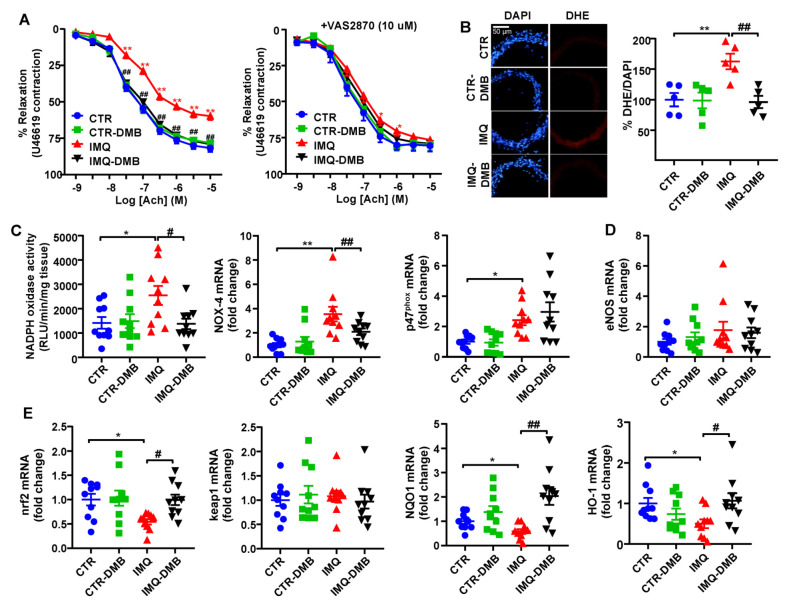
Effects of 3,3-dimethyl-1-butanol (DMB) treatment on SLE-linked endothelial dysfunction, NADPH oxidase activity, and nrf2 pathway in imiquimod (IMQ) mice. (**A**) Acetylcholine (Ach)-induced relaxation responses in aortas pre-contracted by U46619 (10 nM), with or without the specific pan-NOX inhibitor VAS2870 (10 µM). (**B**) ROS content measured by the ratio between red DHE and blue DAPI fluorescence. (**C**) Aortic NADPH oxidase activity determined using a lucigenin-enhanced chemiluminescence and aortic mRNA levels of NADPH oxidase subunits NOX-4 and p47^phox^, (**D**) eNOS, and (**E**) nrf2 pathway, measured by RT-PCR. Groups: control (CTR), CTR-group treated with DMB (CTR-DMB), IMQ, and IMQ-group treated with DMB (IMQ-DMB). The data are shown in a means ±SEM structure. The concentration-response curves to Ach were analyzed by two-way ANOVA with the Tukey’s multiple comparison test. One-way ANOVA and Tukey’s post hoc test or Kruskal–Wallis test with Dunn’s multiple comparisons were performed. * *p* < 0.05 and ** *p* < 0.01 when compared to CTR; ^#^ *p* < 0.05 and ^##^ *p* < 0.01 when compared to IMQ.

**Figure 6 antioxidants-11-00084-f006:**
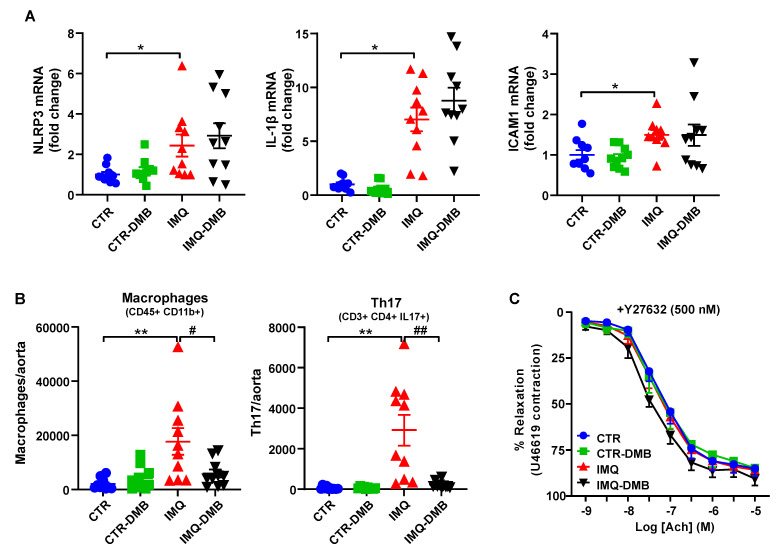
Role of NLRP3 pathway and immune cells infiltration in TMAO-induced endothelial dysfunction in imiquimod (IMQ) animals. (**A**) Aortic mRNA levels of NLRP3 and downstream IL-1β and ICAM1 measured by RT-PCR. (**B**) Th17 and Th1 cells as detected with flow cytometry in aorta. (**C**) Acetylcholine (Ach)-induced relaxation responses in aortas pre-contracted by U46619 (10 nM), with or without the specific Rho kinase inhibitor Y27632 (500 nM). Groups: control (CTR), CTR-group treated with 3,3-dimethyl-1-butanol (DMB) (CTR-DMB), IMQ, and IMQ-group treated with DMB (IMQ-DMB). The data are represented as means ± SEM. The concentration–response curves to Ach were analyzed by two-way ANOVA with the Tukey’s multiple comparison test. One-way ANOVA and Tukey’s multiple comparison test were performed. * *p* < 0.05 and ** *p* < 0.01 when compared to CTR; ^#^ *p* < 0.05 and ^##^ *p* < 0.01 when compared to IMQ.

## Data Availability

The data presented in this study are available in this manuscript.

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
