# Peer review of "Trimethylamine N-Oxide Promotes Autoimmunity and a Loss of Vascular Function in Toll-like Receptor 7-Driven Lupus Mice"

_antioxidants, 2021, doi:10.3390/antiox11010084_

Round 1
Reviewer 1 Report
In this manuscript entitled “Trimethylamine N-oxide promotes autoimmunity and a loss of vascular function in Toll-like receptor 7-driven lupus mice”, the authors evaluated whether trimethylamine N-oxide (TMAO) is associated with the development of autoimmunity in an experimental model of lupus by using trimethylamine lyase inhibitor 3,3-dimethyl-1-butanol (DMB). Given that Toll-like receptor signaling is reported to be important in the pathogenesis of lupus, the results from this study are interesting and adds to our understanding of disease.
I have several comments:
- i) In the 2nd paragraph of Introduction, it is written: “Our group already showed that”
In the 3rd paragraph of Introduction, it is written: “We have already shown that”
-> I would recommend the authors to avoid similar expressions.
iii) In the 4th paragraph of Introduction: “DMB may serve as a therapeutic alternative for SLE.”
-> This sentence could be removed.
- ii) 2.2. Blood pressure, morphology and organ weight indices section:
-> Provide full-terminology for SBP.
-> How was tibia length measured? A detailed description of this procedure will be helpful.
iii) 2.3. Plasma and urine parameters section:
-> It is written anti-dsDNA antibody concentration in the Methods section, while results are
provided as anti-dsDNA IgG activity throughout the figures, which should be clarified.
- iv) 2.6. Flow cytometry section:
-> Provide full-terminology for MLN.
- v) 2.7. Gene expression analysis section:
-> What was the primer sequence for β–actin?
-> Was normalization done with GAPDH or β–actin?
-> qPCR result of RPL13a does not appear in the manuscript. It could be removed if it was not
analyzed.
- vi) 3.1. DMB prevented high blood pressure, target organs damage, and proteinuria in TLR7-
dependent SLE section:
â‘ “Our group assessed the renal function using proteinuria measurements.” -> This sentence
could be removed.
â‘¡ The readership will be particularly interested whether there is a difference in renal
function (creatinine level), as well as the amount of proteinuria after week 4 (Figure 1C).
I would recommend the authors to add creatinine values, if possible.
â‘¢ Were there any mortality in the 4 groups (total n=40)? It was shown that these mice start to
suffer death after 8 weeks of treatment (Arthritis Rheumatol. 2014, 66, 694-706).
vii) 3.3. Plasma TMAO increases in TLR7-dependent SLE and is directly related to systolic
blood pressure and disease activity section:
-> To this reviewer, it is unclear whether it is correct to state “TMAO is directly related to SBP
and disease activity”. Although the authors performed regression analysis, the values of
univariable analysis is identical with Pearson’s coefficient value. I would recommend the
authors to revise the phrase of “directly related to” into “associated with”.
viii) Discussion section:
â‘ In the first paragraph: As the authors stated, the alteration of gut microbiota could be also
responsible regarding to link between TMAO and lupus autoimmunity. Do the authors
have any experimental data for this hypothesis?
â‘¡ High expression of interferon (IFN) signature is a characteristic feature of lupus, and is also
demonstrated in this inducible lupus model (Arthritis Rheumatol. 2014, 66, 694-706).
Did the authors evaluate whether there are changes in IFN signature in the IMQ + DMB
treated group?
â‘¢ In reference 30, plasma TMAO level between SLE patients and mice was not compared,
which should be revised as appropriate.
- ix) Conclusion section:
-> The phrase of “and possibly by controlling choline intake.” could be removed.
Minor comments:
- i) Do plasma TMA and TMAO correlate well? According to Figure 3A, it appears that they are
strongly correlated.
- ii) Please review the manuscript and correct minor typos.
Reviewer 2 Report
In this manuscript, Cristina González-Correa and colleagues show that the systemic lupus erythematosus (SLE) mouse model consisting in the activation of the Toll-like receptor (TLR)7 by imiquimod (IMQ) exhibits an increase in TMAO. In addition, the gut bacterial TMA liase inhibition using trimethylamine lyase inhibitor 3,3-dimethyl-1-butanol (DMB) reduces plasma TMAO levels in this mouse model and alleviates certain clinical symptoms such as proteinuria, auto-antibody production and vascular disfunction. The authors also show that DMB reduces the quantity of Th17 and Th1 cells in this mouse model. Beyond its role on the immune system, the bacterial-derived TMAO contributes to the endothelial dysfunction in SLE mice by the over-production of ROS. The authors claim that DMB improves endothelial function by restoring the nrf2 response and down-stream antioxidant enxymes in this lupus-like mouse model. Although this manuscript is of interest to better understand the IMQ-induced lupus like mouse model, some questions remain to be addressed.
Major points:
1. “In the present study increased plasma levels of anti-dsDNA (Figure 2A), spleen weight/tibia length (Figure 2B), and liver weight/tibia length (Figure 2C) were found in IMQ mice. Interestingly, DMB treatment prevented the increased plasma anti-dsDNA autoantibodies (≈ 35%), splenomegaly(≈ 15%) and hepatomegaly (≈ 23%) in IMQ mice.”
The DMB effect is minor on lymphoproliferation, as observed in figure 2A/B/C. The authors have to rephrase this sentence to account of this minor effect.
2. Figure 2D: could the authors show not only the B-cell population but also show T cell population (CD4+ vs CD8+) ?
3. It is surprising that DMB treatment does not exert a stronger inhibition on TMAO production (see figure 3A), how do the authors explain this ? Could DMB inhibits a different molecular target?
4. Is the observed correlation between TMAO and SBP or anti-DNA in figure 3B is conserved when the correlation is performed with data from IMQ-DMB mice? Indeed, DMB does not seem to efficiently inhibit TMAO production and despite this, DMB treatment dramatically reduces proteinuria and SBP (Figures 1A and 1C). Therefore, this correlation could be lost in IMQ-DMB. How could the authors explain it ?
5. “Vascular infiltration of Th17 is an important element to consider when looking for subjacent mechanisms responsible for the endothelial dysfunction triggered by IMQ microbiota [23]. We found higher macrophage and Th17 accumulation in aorta from IMQ than in CTR mice, which were decreased by DMB treatment (Figure 6B), suggesting that TMAO increased vascular immune cells infiltration.”
Th17 recruitment in kidneys has been associated with the presence of a chemoattractant, CD95L in SLE mice 1. Could the authors check the level of this cytokine in their mouse model ?
reference
1. Poissonnier A, Sanseau D, Le Gallo M, et al. CD95-Mediated Calcium Signaling Promotes T Helper 17 Trafficking to Inflamed Organs in Lupus-Prone Mice. Immunity. 2016;45(1):209-223.
Author Response
We thank the reviewer for the helpful comments and the positive criticisms. Following his/her suggestions we have made changes to the text, which, we believe, have improved the manuscript.
Major points:
Point 1: “In the present study increased plasma levels of anti-dsDNA (Figure 2A), spleen weight/tibia length (Figure 2B), and liver weight/tibia length (Figure 2C) were found in IMQ mice. Interestingly, DMB treatment prevented the increased plasma anti-dsDNA autoantibodies (≈ 35%), splenomegaly (≈ 15%) and hepatomegaly (≈ 23%) in IMQ mice.”
The DMB effect is minor on lymphoproliferation, as observed in figure 2A/B/C. The authors have to rephrase this sentence to account of this minor effect.
Response 1: Following your suggestion, we rephrase this sentence: “Interestingly, DMB treatment partially prevented the increased …”
Point 2: Figure 2D: could the authors show not only the B-cell population but also show T cell population (CD4+ vs CD8+) ?
Response 2: Following your suggestion we show total T cells (CD45+ CD3+) and Th cells (CD45+ CD3+ CD4+). In our experiment we did not measure CD8+ cells.
Point 3: It is surprising that DMB treatment does not exert a stronger inhibition on TMAO production (see figure 3A), how do the authors explain this ? Could DMB inhibits a different molecular target?
Response 3: DMB inhibits some, but not all, microbial TMA lyases (Wang et al., 2015, Cell 163, 1585–1595). Thus, it is not surprising that DMB treatment did not exert a stronger inhibition on TMAO production. Moreover, DMB promoted reduction in proportions of some microbial taxa that are associated with plasma TMA and TMAO levels (Wang et al., 2015, Cell 163, 1585–1595).
Point 4: Is the observed correlation between TMAO and SBP or anti-DNA in figure 3B is conserved when the correlation is performed with data from IMQ-DMB mice? Indeed, DMB does not seem to efficiently inhibit TMAO production and despite this, DMB treatment dramatically reduces proteinuria and SBP (Figures 1A and 1C). Therefore, this correlation could be lost in IMQ-DMB. How could the authors explain it?
Response 4: The correlation analysis presented in Figure 3B was performed with data from all experimental groups, including IMQ-DMB group.
Point 5: “Vascular infiltration of Th17 is an important element to consider when looking for subjacent mechanisms responsible for the endothelial dysfunction triggered by IMQ microbiota [23]. We found higher macrophage and Th17 accumulation in aorta from IMQ than in CTR mice, which were decreased by DMB treatment (Figure 6B), suggesting that TMAO increased vascular immune cells infiltration.”
Th17 recruitment in kidneys has been associated with the presence of a chemoattractant, CD95L in SLE mice 1. Could the authors check the level of this cytokine in their mouse model?
reference
1. Poissonnier A, Sanseau D, Le Gallo M, et al. CD95-Mediated Calcium Signaling Promotes T Helper 17 Trafficking to Inflamed Organs in Lupus-Prone Mice. Immunity. 2016;45(1):209-223.
Response 5: I agree with the reviewer on the relevance of knowledge of CD95L changes in SLE. In fact, there is no information that TMAO can alter the expression of this cytokine. Our data suggest that plasma TMAO levels found in IMQ mice were not sufficient to induce direct action at the vascular tissue. However, regardless of the mechanism, we believe that it is very relevant that we found reduced Th17 infiltration in aorta when plasma TMAO levels were reduced by DMB treatment. Whether CD95L is involved in TMAO-induced Th17 infiltration of vascular tissue in this lupus model requires a thorough investigation that is beyond the scope of the present work.
Round 2
Reviewer 1 Report
Dear authors,
Thank you for the revision.
Reviewer 2 Report
The authors addressed most of my concerns.